# ModE-Sim - A medium size AGCM ensemble to study climate variability during the modern era (1420 to 2009)

Ralf Hand[1,2], Eric Samakinwa[1,2], Laura Lipfert[1,2], and Stefan Brönnimann[1,2]

[1]Institute of Geography, University of Bern, Switzerland.
[2]Oeschger Centre for Climate Change Research, University of Bern, Switzerland.

**Correspondence:** Ralf Hand (ralf.hand@giub.unibe.ch)

**Abstract.**

We introduce ModE-Sim, a medium size ensemble of simulations with the atmospheric general circulation model ECHAM6 in its LR version (T63/approx. 1.8°horizontal with 47 vertical levels). The ensemble uses prescribed sea surface temperatures, sea ice and radiative forcings that reflect observed values while accounting in uncertainties in the ocean conditions and the timing and strength of volcanic eruptions. The simulations cover the period from 1420 to 2009. With 60 ensemble members between 1420 and 1850 and 36 ensemble members from 1850 to 2009 ModE-Sim consists of 31620 simulated years in total. ModE-Sim is suitable for many applications as its various subsets can be used as initial condition and boundary condition ensemble to study climate variability. The main intention of this paper is to give a comprehensive description of the experimental setup of ModE-Sim and to provide an evaluation of the two key variables 2m-temperature and precipitation. We demonstrate ModE-Sim's ability to represent their mean state, to produce a reasonable response to external forcings and to sample internal variability. At the example of heat waves we show that the ensemble is even capable of capturing certain types of extreme events.

# 1  Introduction

The use of large ensembles of climate model simulations enables to separate the climate's response to external forcings from internal climate variability (Maher et al., 2019; Milinski et al., 2020; von Trentini et al., 2020). The individual realizations of a single-model ensemble can differ either in their boundary conditions, their initial conditions, or in both.

If the ensemble size is large enough and capable of spanning the full range of physically plausible climate states, large ensembles are likely to include realizations that are close to the historically observed climate state within a reasonalbe range of uncertainty.

Some notable applications include but are not limited to the analysis of Sea Surface Temperature (SST) trends (Olonscheck et al., 2020), the production of initial state estimates for reanalyses data sets via data assimilation approaches (Bhend et al., 2012; Franke et al., 2017; Valler et al., 2022), modulation of global warming (Liguori et al., 2020) and extreme events (Landrum and Holland, 2020).

Before the widespread use of large ensembles, separating internal variability and external components of climate variability in model simulations usually was done by comparision of the statistics of a transient simulation with the statistics of a control simulation with climatological forcings. This approach does not fully allow inferences about the internal variability of such simulations, as it has limitations to capture changes of internal variability over time (Maher et al., 2015, 2019). To this end, large ensembles provide a more accurate method of separating the different components of climate variability and account for the changes over time, but are computationally demanding. With the increasing number of available resources, Single Model Initial-condition Large Ensembles (SMILEs hereafter) have become an important tool for understanding climate variability (e.g., Deser et al., 2020; Maher et al., 2021). Furthermore, large ensembles offer the opportunity to investigate the range of climate variability in a physically consistent framework designed to offer a representative sample of the range of climate states under the given forcing.

Here we present Modern Era SIMulations (ModE-Sim), a medium-sized ensemble of simulations with an atmospheric model capturing the period 120 to 2009 (later refered to as the "modern era"). Unlike some notable multi-model ensemble simulations offered by model intercomparison projects, our setup is similar to a SMILEs setup using a single model but additionally accounts for uncertainties in the boundary conditions. We use the atmospheric general circulation model ECHAM6 with forcings that account for uncertainties in the SSTs, sea ice, and the effect of volcanic eruptions. The experimental design of most SMILE simulations depends on its purpose (Maher et al., 2021). ModE-Sim aims to provide an initial state estimate for ModE-RA, a paleoclimate data assimilation approach, thereby producing a 3-dimensional reconstruction of the global climate for the according modern era (Valler et al., 2023, submitted). Focusing on the key variables necessary for our data assimilation procedure, namely surface air temperature and precipitation, we evaluate how well ModE-Sim represents the mean observed climate state and variability during the time for which reliable observations are available. We show that our ensemble size is sufficient in sampling the internal variability of the key variables and is also capable of capturing extreme events, such as heat

waves. Providing boundary conditions for the early part of the modern era is challenging but necessary. Our ensemble simulations use HadISST2 sea SSTs and sea ice concentrations (Titchner and Rayner, 2014) in the later periods and SSTs and sea ice concentrations based on an ensemble multi-proxy temperature reconstruction over the modern era (Neukom et al., 2019).

These multi-proxy temperature reconstructions are selections based on the agreement between marine proxies and simulated SST from the *past1000* of PMIP3 (Gómez-Navarro et al., 2017), resulting in physically consistent ensemble fields (Samakinwa et al., 2021).

We organize the remaining parts of this work as follows: In section 2, we describe the initialization and experimental design procedure of ModE-Sim. Section 3 give explanations of the statistical methods used for the evaluation of the ensemble

simulations while section 4 concludes the manuscript and provides a summary.

## 2 Model & experimental setup

### 2.1 The model: ECHAM6

We use ECHAM6, the atmospheric component of the Max Planck Institute's Earth System Model (MPI-ESM), for all our simulations. MPI-ESM and ECHAM6 stand-alone models participate regularly in the Atmospheric/Coupled Model Intercom-

60 parison Project (AMIP/CMIP). We will only briefly describe the configuration used in this study, for further details on the model, please refer to Stevens et al. (2013). We use ECHAM version 6.3.5p2, the CMIP6 version, and presumably the final development step of the ECHAM model family. Furthermore, we use the low resolution (LR) version of the model, with a horizontal resolution of T63 equivalent to a grid width of approximately 1.8°. In the vertical, the LR version uses 47 hybrid levels between the surface and 0.01 hPa.

### 2.2 The experiments

In principle, our set-up is designed to be close to the PMIP4 *past2k* simulations performed at the Max Planck Institute for Meteorology (Jungclaus et al., 2017) with few exceptions. Our simulations comprise three epochs, forced with slightly different ocean boundaries and radiative forcings. Figure 1 gives an overview of our experimental design.

As a main difference, in contrast to the past 2k simulations that use the coupled version of the model, here we use the

70 stand-alone atmospheric component of the model with prescribed SST and sea ice. This setup induces that the ensemble spread does reflect internal variability in the ocean, but ties the ocean to observed conditions. However, as there is high uncertainty in the SSTs, particularly in the early period, we account for these uncertainties by using ensembles of SSTs with an individual realization of the SST making the ocean forcing for each ensemble member of our simulations. In principle, earlier forcings include larger uncertainties. We account for the latter by choosing a larger ensemble size and a wider variety of forcings for

the period prior to 1850. While date and strenght of volcanic eruptions are relatively well constrained by observations in the time after 1850, there is more uncertainty in the prior period. We account for this fact by varying the volcanic forcings in one of our subsets for the time prior to 1850.

As another difference to the past2k simulations, we use prescribed land-cover maps rather than dynamic vegetation.

The initialization of our model simulation is in two steps: First, we forked an atmosphere-only spin-up simulation from
the PMIP4 simulation of the coupled version of the model (i.e. MPI-ESM). This spin-up simulation was forced with constant
boundary conditions and radiative forcings spanning several decades until equilibrium is reached in an atmosphere-only mode.
Secondly, we then forked the actual transient simulations from different time instances of this spin-up run.

We performed two spin-up simulations to generate a set of initial conditions for our transient simulations. The transient sim-
ulations include 36 ensemble members initialized in the year 1850 and 60 ensemble members initialized from 1420 conditions.
We account for uncertainties of the lower boundary conditions by using different realizations of SSTs from the HadISST2
dataset for the runs starting in 1850 and an ensemble of novel SST reconstructions (Samakinwa et al., 2021) for the simulations
from 1420. Furthermore, our setup accounts for uncertainties in the radiative forcing.

### 2.2.1   Spin-up runs and initialization strategy

Due to deficiencies in the representation of the large-scale ocean circulation in MPI-ESM, the ocean boundary conditions pro-
vided by the ocean component of the coupled system show distinct differences to observed SSTs in some regions (Müller et al.,
2018, Fig. 2a therein). Therefore, an abrupt switch from MPI-ESM (coupled system) to the atmosphere-only setup might cause
an initial shock to the atmospheric circulation.

For our transient runs, we performed two spin-up simulations initialized from the years 1420 and 1850 of the coupled PMIP4
simulations. These enable a smooth transition from the PMIP4 coupled simulations to our atmosphere-only ensemble. It also
allows the use of slightly different initial conditions for the individual members. The spin-up runs use forcings and boundary
conditions of the initialization year (monthly-varying, but no year-to-year variability) such that each year of the spin-up run is a
realization of the years 1420 and 1850, respectively. A potential caveat of this strategy is that we start our transient simulations
from a world that has not experienced inter-annual variability of the forcing for several years. However, a realistic state of the
atmosphere is achieved after a few years due to its short memory.

### 2.2.2   Transient simulations epoch 1: 1420 to 1850

The simulations of epoch-1 consist of 60 simulations, divided into three subsets. The first subset (**"set 1420-1"**) consists of 20
simulations that were forced with 20 different ocean boundary conditions. The procedure used to generate the oceanic bound-
ary conditions is described in Samakinwa et al. (2021). We initialize from 20 different time instances of the 1420 spin-up run.
All these 20 simulations share the same radiative forcing that is identical to the standard PMIP4 setup.

The second subset (**"set 1420-2"**) consists of another 20 simulations that use another 20 realizations of the SST reconstruc-
tions and another 20 initializations. The difference to set 1 is in the radiative forcing: In contrast, the simulations in set 2 each
have a different volcanic forcing. Consistent with the PMIP4 standard setup, these radiative forcings are outputs of the Easy

Volcanic Aerosol Model (Toohey et al., 2016, EVA) using Volcanic Stratospheric Sulfur Injections reconstruction of Sigl et al. (2022). PMIP4 volcanic forcing also results from EVA, generated using VSSI reconstruction of Toohey and Sigl (2017). The individual realizations of the volcanic forcing account for uncertainties in the timing and strength of the eruptions by varying the according quantities in the EVA input. Both sets 1420-1 and 1420-2 use sea ice analogs selected from the HadISST2 dataset based on a pattern-matching algorithm applied to the SSTs (For details, please refer to Samakinwa et al. (2021)).

The last subset (**"set 1420-3"**) uses the same 20 initializations as set 1420-1. While sets 1420-1 and 1420-2 use a preliminary version of the SST reconstructions, set 1420-3 utilizes the final version. Due to slight modifications in the algorithm that generates the SST reconstructions, the SST variability is reduced by up to approximately 20% in set 1420-3 w.r.t. the first two sets of epoch 1 (see Fig. 3). A first analysis shows that the effect of this reduction seems to have a minor influence on the surface temperature variability over land, but is limited to the affected ocean grid points. Another difference to the previous sets is that set 1420-3 uses HadISST2 historical sea ice climatology rather than the sea ice analog approach used for sets 1420-1 and 1420-2. We found that the sea ice analogs approach shows a bias as almost all analogs were sampled from the late 20[th] century. This results in low sea ice concentration, particularly ice-free conditions in the marginal seas like the Labrador Sea and Sea of Okhotsk, which is unlikely to be realistic for the earlier periods. The effect of this seems to be limited to the direct surrounding where the sea ice cover changed, but simulations with climatological sea ice might be more suitable for certain analyses. Note, that there are hardly any observations for the high latitudes prior to 1970, so any information on historical sea ice is subject to very high uncertainty. Our different sets can therefore be seen as the upper and lower bounds of a very conservative uncertainty range. Starting in 1780 ("epoch 1b"), the SST reconstruction assimilates large bodies of marine observations, pulling the ocean towards its true state. Because the differences between the SST/Sea ice forcing between epoch 1a and epoch 1b are small while uncertainty in the pre-1780 forcing is high, we did not perform an additional spin-up for the epoch 1b simulations but directly continued these runs from the epoch 1a simulations with only swapping to the new forcing.

### 2.2.3 Epoch 2: 1850 to 2009

From 1850 onwards ("epoch 2"), uncertainties in the forcings become smaller. Most major volcanic eruptions are well documented and ocean boundary conditions are more and more constrained by ship measurements and – from the end of the 20[th] century onwards – also satellite measurements. For epoch 2 our simulations consist of two subsets that differ only slightly in their ocean boundary forcings and their initial conditions. **Set 1850-1** was started from 20 different time instances of the spin-up simulation and uses 10 different realizations of HadISST2 as ocean boundary conditions (i.e. each HadISST realization is used with two different initializations). To expand the sample of SST forcings the second set (**"set 1850-2"**) uses 16 different recombinations of the 10 available HadISST2 realizations as ocean forcing. These recombinations were formed as follows:

$$SST(i) = (SST(j)/SST(k)) * SST(l), \quad with \quad j \neq k \neq l$$

where $SST(i,x,t)$ is the SST anomaly in the i[th] recombination, and $SST(j)$, $SST(k)$ and $SST(l)$ are the according SST anomalies in the j[th], k[th] and l[th] realization of HadISST2. The sea ice concentrations were created by choosing analogs from the

HadISST2 sea ice data set. The radiative forcings, including volcanic forcings, are the PMIP4 input and are identical for all simulations and all sets of epoch 2.

## 3  Evaluation

To make an ensemble useful for studying climate variability, it is required to have information on how well its setup represents the mean state of the variables of interest and to what extent the ensemble can sample the forced and internal variability of the atmosphere. The following section will provide some analysis of these questions, focusing on the variables surface temperature and precipitation, as these are the variables of highest interest for the data assimilation in our project. Concerning variability, we consider different time scales, including annual, seasonal, and monthly variability. As reference data set to compare with, we mainly use the Berkeley Earth dataset (Rohde and Hausfather, 2020), as it goes back to 1750 and therefore allows evaluation of our pre-industrial simulations. We evaluate two periods, 1780 to 1850 and 1950 to 2000. For precipitation, we use the Global Precipitation Climatology Centre (GPCC) dataset (Becker et al., 2013). We limit the evaluation to the period from 1950 to 2000 because there are no reliable global precipitation data sets available for the earlier period.

### 3.1  Evaluation of mean state biases of precipitation and 2m temperature

A detailed analysis of mean state biases of ECHAM6 is beyond the scope of this paper. For a more detailed mean state analysis, please refer to Stevens et al. (2013) and Giorgetta et al. (2013). ModE-Sim reproduces the known anomalies of ECHAM6 when used in AMIP mode (suppl. fig. S1). The temperature bias simulated for epoch 1b has the same spatial extent as for the period between 1950 and 2000. The main features are a warm bias in the northern hemisphere mid-latitudes and over Australia and cold biases over South America, India, and the northern Rocky Mountains. We also found a wet bias over the Himalayas and the Andes. Our results are in agreement with the existing studies (Giorgetta et al., 2013).

### 3.2  Response to the external forcings

ModE-Sim reproduces the observed global mean near-surface temperature (Fig. 2). The time series shows a clear cooling imprint following volcanic eruptions and the warming trend in the 20[th] century. The forced signal related to the radiative forcing and the ocean boundary conditions can be detected from the subensemble means computed from each 20-member set separately (Fig. 2b & c), indicating that the ensemble size is clearly sufficient to separate forced signals from internal variability of the atmosphere and uncertainties related to forcing and boundary conditions. The ensemble spread declines towards the end of the simulations, indicating that for the earlier period, the forcing/boundary condition uncertainty contributes to the ensemble spread in the same order of magnitude as internal variability does.

To determine the spacial manifestation of the volcanic signal we computed composites of the ensemble mean anomalies of 2m temperature, precipitation and sea level pressure for the first (fig. 4) and second (suppl. fig. S3) winter and summer after 15 maior eruptions (Fischer et al., 2007, table 1 therein) with the 5 summers/winters prior to each of the eruptions as reference period. To test the response for statistical significance we have additionally created 1000 surrogates with each surrogate being

created by picking 15 random years (with replacement) and computing the difference between these 15 random years and the according 5-year periods prior to them. We defined those parts of the response as statistically significant that falls outside the $5^{th}$ to $95^{th}$ percentile range of these 1000 surrogates.

The spatial anomalies agree well with observations and previous modelling studies (e.g., Graft et al., 1993; Robock, 2000; Fischer et al., 2007; Sjolte et al., 2021). The most prominent feature in terms of 2m temperature is a direct response to the negative radiative forcing that resembles a strong cooling over most continental regions that regionally exceeds 1K and is statistically significant for most regions. The most prominent exception from this cooling response can be found over northern Eurasia in boreal winter, but which is only statistically significant in its center. The underlying mechanisms of this winter warming is not fully understood yet and differs between different climate models (Driscoll et al., 2012) and forcing datasets (Zambri et al., 2017). Based on analysis of simulations with MPI-ESM, the coupled version of ECHAM6, Bittner et al. (2016) discuss that the warming may be caused by changes in midlatitude lower stratosphere zonal winds that lead to an equatorward deflection of planetary waves. This reduces higher latitude wave breaking and hence disturbances of the polar vortex. In ModE-Sim we can find anonamously negative SLP response in the northern polar latitudes in connection with a band of positive SLP anomaly that spans over the North Atlantic and Northern Eurasia. This pattern, also known as the positive phase of the Arctic Oscillation (or North Atlantic Oscillation when restricted only to the Atlantic region) is a known mode of large scale variability in the Northern hemisphere and its excitation in the first winter after volcanic eruptions is supported by observations (Christiansen, 2008; Fischer et al., 2007). A positive AO/NAO in winter typically leads to enhanced advection of marine air to the continent, resulting in the winter warming. While the warming is significant at least in its center, the SLP response itself is not, likely due to the high internal variability in SLP. The most prominent feature of the precipitation response is a shift of the innertroppical convergence zone towards the summer hemisphere that is strongest over the central and West Pacific.

Exemplary for atmospheric response to the ocean boundary conditions we analyzed the response to ENSO in boreal winter (DJF). We computed an ENSO index that basically follows the method provided in Trenberth (1997): We took the monthly deviations of SST from the climatology of epoche 1a, averaged these anomalies for the Nino 3.4 region (5°N to 5°S, 170 °W to 120°W) and applyied a 5-month running mean. Afterwards, we computed regressions of atmospheric surface quantities in boreal winter on this index for all sets of epoche 1a. Due to the large ensemble size, the correlation is found to be highly significant in almost all regions of the world. ModE-Sim is able to reproduce the main features of the known ENSO teleconnection (Fig. 5): The SLP response shows good agreement to other models and reanalysis data (e.g., Döscher et al., 2022, fig 18 therein) with a streghening of the Aleutian low, a weekening of the NAO, negative SLP anomalies in the eastern tropical Pacific and a band in the subpolar Southern Ocean, moderate positive SLP anomalies over the Arctic, the Antarctic and a strongly positive SLP anomaly west over the Amundsen Sea. The observed 2m temperature response (Brönnimann, 2007) is well captured around the Pacific and Europe, with a warming over Northern Australia, a cooling over the US, a warming over Kanada and Alaska connected to the weakened Aleutian Low and cooling over Europe induced by the weakened NAO. Concerning the precipitation response The wettening of the US and central Brazil is captured, as well as the wettening of tropical South America and over Australia and Malaysia.

### 3.3 Differences between the individual sets in epoch 1

For epoch 1, we computed 3 sets that differ in the ocean boundary conditions and the radiative forcing (see section 2.2.2). A resulting question is whether these 3 sets show substantial differences in mean state and ensemble spread. Fig. 2b shows that all sets have similar features in their global mean surface temperature for both externally forced signal and ensemble spread. Set 1420-3 shows an offset towards lower temperatures. These low temperatures are related to the enhanced sea ice extent when forced with climatological sea ice conditions. This offset vanishes when considering land grid points only.

The main differences in the mean state are found in the high latitudes and are the plausible response to the different sea ice forcing. In the lower latitudes, the main features are warm anomalies over subtropical North Africa and along the Brazilian North East coast. These warm anomalies are likely related to a slightly warmer state of the Atlantic in the forcing for set 1420-3. However, the amplitude of these anomalies is small compared to the expected variability of 2m-temperature (suppl. fig. S3). Also, the differences in terms of the SST forcing are relatively small compared to the variations caused by different states of the ocean circulation.

### 3.4 Ability of ModE-Sim to sample internal variability

MPI-ESM, the coupled version of ECHAM6, has been shown to be able to sample internal variability when using a 100-member ensemble of historic simulations (Suarez-Gutierrez et al., 2021). We apply the method proposed in the latter reference to ModE-Sim, to analyze whether the ability of capturing internal variability also holds in a stand-alone atmospheric mode and for our ensemble with fewer ensemble members that extend further into the past. The method has the strength to evaluates internal variability in a model ensemble without making a priori assumptions. A detailed description of it can be found in the reference, briefly summarized it works as follows: First, one calculates the ratio of timesteps where the observations fall outside the ensemble minimum to ensemble maximum range at the same timestep. If the ensemble captures the spread correctly, then such outlayers should only occur very occassionally (how often exactly depends on the ensemble size; e.g. a 20-member ensemble with realistic internal variability should be likely to capture all events with a 20-year return period, so on average in 5 % of the timesteps an observation should fall outside the ensemble minimum to ensemble maximum range). If the observation tend to be only above (only below) the ensemble maximum (minimum) for an overproportional number of timesteps then this indicates a negative (positive) mean state bias of the model. If outlayers w.r.t. the ensemble min-to-max range occur frequently into both, positive and negative direction, this indicates that the model underestimates observed internal variability. Finally, the method also allows to detect regions where the model overestimates internal variability w.r.t. observations. This is the case if an overproportional amount of observations falls within the center of the ensemble spread. Consistent with Suarez-Gutierrez et al. (2021) we here use the 12.5 to 87.5 quantile range, which - if the ensemble spread agrees with observed internal variability - by definition on average should include 75 % of the observations.

For 2m-temperature ModE-Sim performs best over Eurasia and tropical South America, as well as over parts of North America (Fig. 6a - 6d). This performance holds for both monthly and yearly averaged anomalies. Generally, there is a better agreement for the period 1950 to 2005 than for 1780 to 1850. In most other regions, the ensemble spread tends to be too large, indicated by the hatched regions where the observations fall within the 12.5 to 87.5 percentile range in an over-proportional number of time steps. On seasonal timescales, it shows that the performance in boreal winter is better than in the summer sea-

son (suppl. fig. S4). The results for temperature hold even when analyzing each set separately, indicating that only 20 ensemble members already give a reasonably good estimate of internal variability (suppl. fig. S5).

     For precipitation, we limit our analysis to the period 1950 to 2005 because of the lack of precipitation observations before the $20^{th}$ century. During this period, ModE-Sim captures annual precipitation variability reasonably well in most parts of the

250 world (Fig. 6e), but slightly too high ensemble spread is found over parts of Europa and western Siberia and parts of North America. The ensemble spread is also large in Northern Africa, over Eastern Siberia and the Tibetian plateau. On monthly time scales, ModE-Sim shows the tendency to be too wet in many timesteps in the subtropical desert regions (Fig. 6f).

### 3.5   Heat w aves as an example to demonstrate ModE-Sim's ability to simulate extreme events

Beyond its ability to separate internal variability and externally forced events, a large ensemble can also be a useful tool for studying climate extremes. By definition, extreme events are rare and therefore usually only a limited number of observations is available. At a certain number of ensemble members, even extreme events may occur often enough to be able to perform a statistically robust analysis of the underlying mechanisms. Also it is interesting to know to what extent these extremes are influenced by external forcing. A detailed study of extreme events is beyond the scope of this documentation paper, but we

will show some first results that give evidence that ModE-Sim captures heat waves, making it a valuable tool for studying the underlying processes triggering climate extremes.

     For our analysis we computed the number of heatwaves per season for the model simulations and, as a reference, for 20CRv3 and ERA5 reanalysis. Based on daily 2m temperature data we defined heatwaves as those days on which temperature exceeds

the $90^{th}$ percentile of a reference period at the according grid point. We used two different definitions of the reference period: In the first approach the percentiles were computed from the 1961 to 1990 period. This fixed climatology approach has the advantage that the threshold for a day becoming labeled as a heat wave day does not change during time. The obvious disadvantage is that the number of heat wave days is likely to increase with rising mean temperature. Therefore we additionally provide a second version of the analysis where the percentiles are computed from a 31-year running window. This adapts the temperature

threshold for heatwaves to the anthropogenic warming and heatwaves are restricted to days that are extremely warm w.r.t. other days in the according 31-year window. To correct for model biases the thresholds were computed separately for each dataset. To compare the datasets we computed the spearman rank correlation between the model and the reanalysis and according p-values.

Fig. 7 shows that ModE-Sim is able to produce a reasonable number of heatwaves on a hemispheric scale. While the occurrence of heat waves is slightly underestimated for the Northern hemisphere before 1950, the number of heat wave days almost perfectly matches that in 20CRv3 when considering the ensemble mean in the Southern hemisphere. The high correlation between the number of observed heat waves and the ModE-Sim ensemble means indicates that a large part of the heat waves can be linked to external forcing, particularly for the Southern hemisphere. External forcing in this case does not necessarily restrict to a direct radiative response, but may also include dynamical components that are a result of the SST forcing and/or the volcano forcing. High correlations are not only resulting from the strong temperature trend in the late 20[th] century, but also hold with the 31-year moving climatology approach (lower row of Fig. 7). Discrepancies with 20CR in the Southern hemisphere in the early 20[th] century are likely a result of missing data and high uncertainties in 20CR during that period.

A more detailed study of heat waves in ModE-Sim is currently in preparation.

## 4  Conclusions

In this paper, we presented ModE-Sim, a medium-sized ensemble of AGCM simulations covering the period from 1420 to 2009. To our knowledge, it is the first ensemble of comparable size that covers such a long period. ModE-Sim has to be seen as an ensemble of opportunities that combines different, partly inhomogeneous setups. These reflect the very different levels of uncertainty in the forcing and boundary conditions for the different periods. Due to the re-initialization of the model in 1850, the switch from SST reconstructions to HadISST forcing, and the change in ensemble size, we strongly recommend analyzing epoch1 and epoch 2 separately when using the dataset. Except for localized effects in the high latitudes when switching between climatological sea ice forcing and analogs, we only found minor differences between the individual sets in epoch 1. Therefore we can treat all three sets as one 60 member-ensemble when the data is used for analysis that benefits from a large ensemble size. However, the more conservative and accurate way would be to analyze each set separately.

Nevertheless, we show that ModE-Sim is a useful tool to study climate and its variability in the past 600 years. While sharing the previously known mean state biases of ECHAM in stand-alone mode, our ensemble performs well in sampling internal variability, particularly for near-surface temperature. Another interesting finding is that the ModE-Sim has the ability to capture extreme events, such as heat waves.

Beside its usefulness for pure model studies the original motivation for ModE-Sim was to create the input for an offline data assimilation approach. The according reanalysis product, ModE-RA (Valler et al., 2023, submitted, data available at https://www.wdc-climate.de/ui/entry?acronym=ModE-RA), is published in a consistent data structure to easily allow direct comparision between the AGCM ensemble and the climate reconstruction based on it.

*Code availability.* ECHAM6 was published by the Max Plack Institute for Meteorology (MPIMET) under an institutional licence that guarantees access to the ECHAM6 source code to the scientific community. Accessing the ECHAM6 source code requires contacting the modelling department of MPIMET: https://mpimet.mpg.de/en/research/modelling. Additionally, for documention of the ECHAM6 setup that was used to create ModE-Sim, ECHAM6 example run scripts are provided as additional information with the data through the World Data Center for Climate: https://www.wdc-climate.de/ui/entry?acronym=ModE-Sim. Code used to create EVA inputs and generate perturbed
volcanic forcing can be found here: https://doi.org/10.5281/zenodo.7669569.

*Data availability.* A subset of ModE-Sim variables (including, but not limited to, these used in this manuscript), forcings and boundary conditions (unless standard PMIP4/HadISST2) and example run scripts are made available through the World Data Center for Climate: https://www.wdc-climate.de/ui/entry?acronym=ModE-Sim. Individual ModE-Sim sets also can be accessed through the following DOIs: https://doi.org/10.26050/WDCC/ModE-Sim_s14201 (Set 1420-1), https://doi.org/10.26050/WDCC/ModE-Sim_s14202 (Set 1420-2),
https://doi.org/10.26050/WDCC/ModE-Sim_s14203 (Set 1420-3), https://doi.org/10.26050/WDCC/ModE-Sim_s18501 (Set 1850-1), https://doi.org/10.26050/WDCC/ModE-Sim_s18502 (Set 1850-2). Further variables are avaiable upon request by contacting the authors.

*Author contributions.* Ralf Hand was leading the writing of the manuscript, performed the AGCM experiments and did most of the presented evaluation. Eric Samakinwa provided the ocean boundary coundisions and generated the perturbed volcanic forcing. Laura Lipfert did the analysis on the heat waves part. All authors contributed to the discussion on the experimental setup, the analysis and the manuscript.

*Competing interests.* None

*Acknowledgements.* We would like to thank Doris Folini, Sylvaine Ferrachat and the climate modeling group at ETH Zurich and Johann Jungclaus, Sebastian Rast and their colleagues at the Max Planck Institute for Meteorology for assistance with installing the model and the selection and configuration of the forcings. We acknowledge technical support by the Swiss National Supercomputing Center (CSCS) and the University of Bern HPC team. Furthermore we would like to thank Eileen Hertwig, Heinke Höck and the data management team at the
325 German Climate Computing Centre (DKRZ) for comprehensive support with publishing the datasets through the WDCC platform. RH, ES and LL were funded by the European Research Council through H2020 (ERC Grant PALAEO-RA 787574).

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

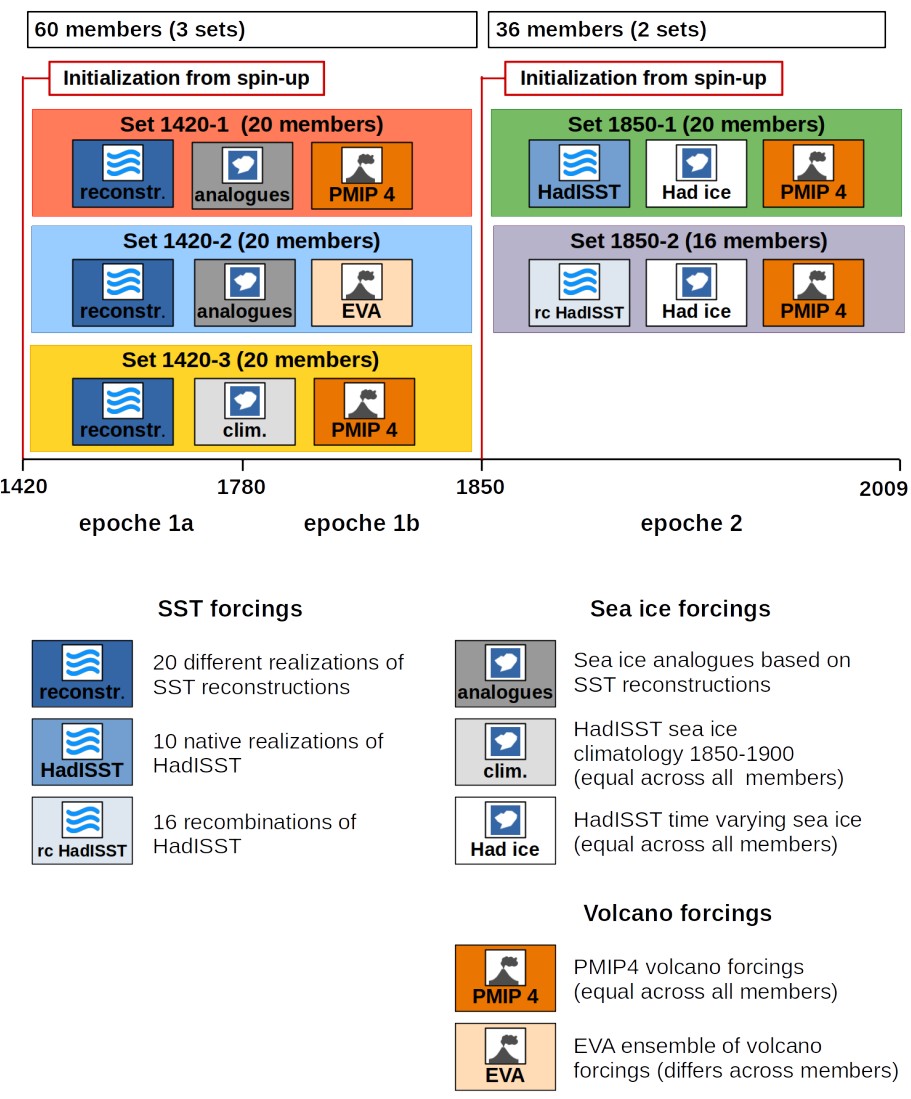

**Figure 1.** Overview on the setup of the experiment sets in ModE-Sim and their forcings.

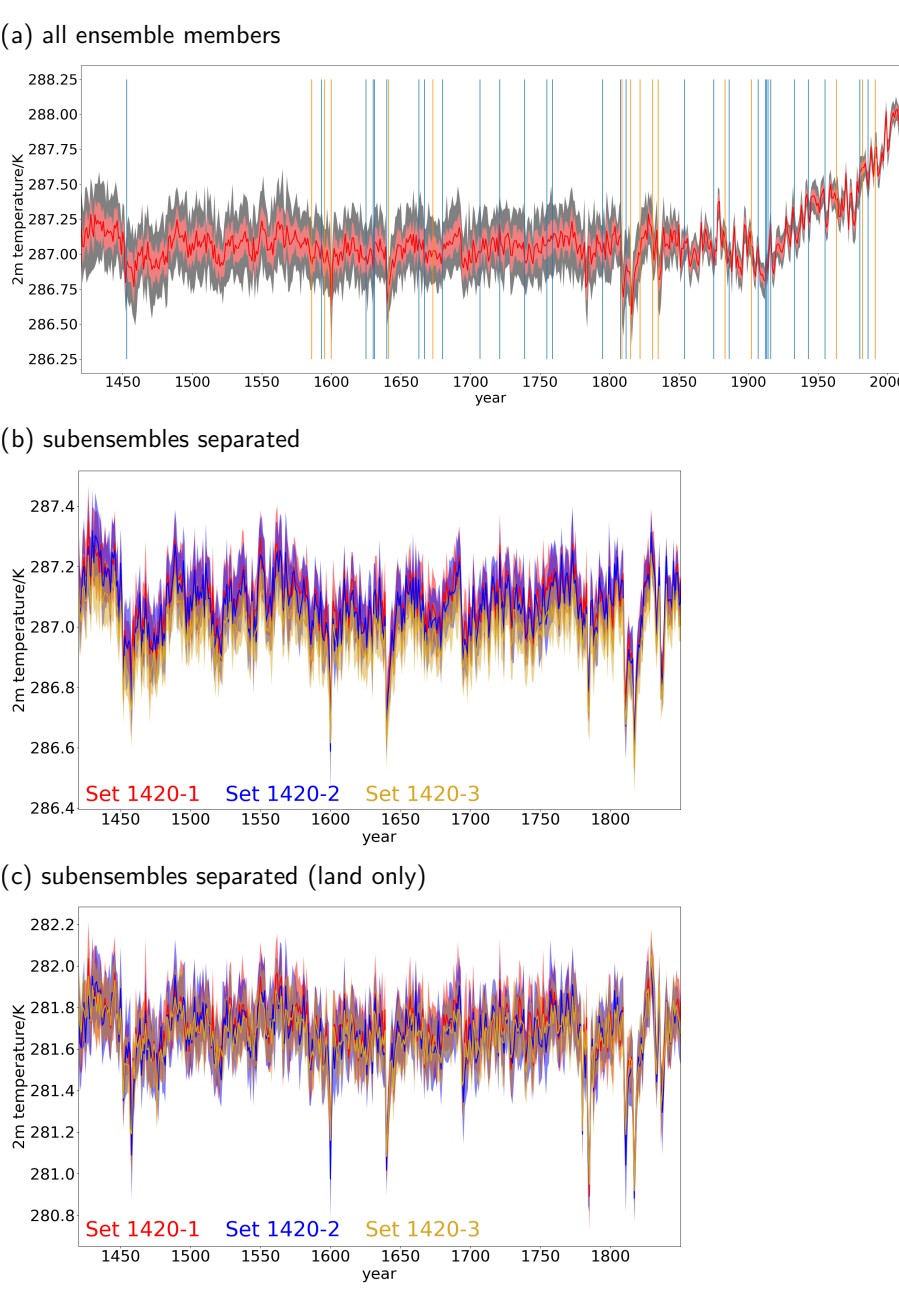

**Figure 2.** Time series of globally averaged 2m temperature. (a) Ensemble mean (red line), +/- 1 ensemble standard deviation (light red shading), and ensemble minimum/maximum (grey shading) of all sets in epoch 1. Vertical orange lines indicate volcanoes used for the composites in section 3.2, vertical blue lines indicate additional volcanic eruptions with a volcanic explosivity index $\geq 5$. (b) Sets 1420-1 (red), 1420-2 (blue) and 1420-3 (dark yellow) separated. The coloured lines indicate the ensemble mean of each set, the shadings the ensemble minimum-maximum range. (c) same as (b), but limited to grid points over land.

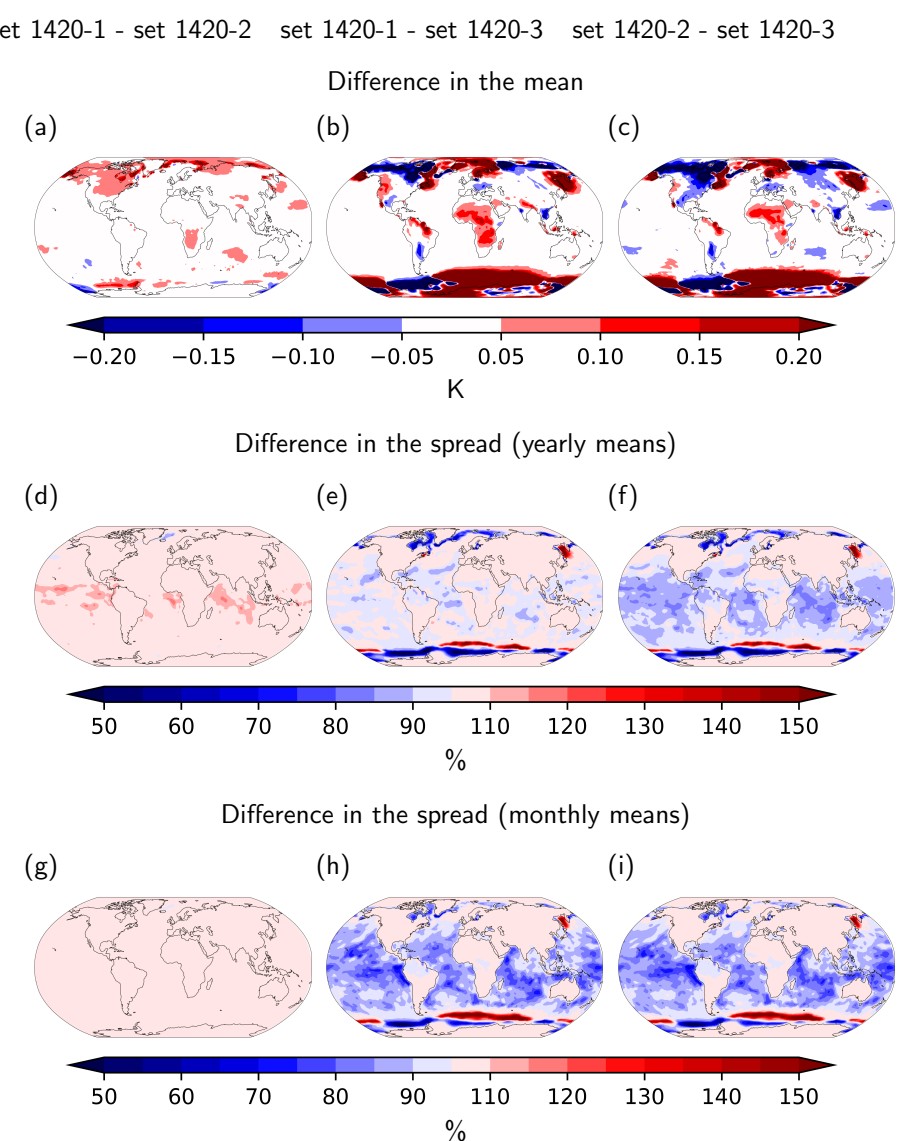

**Figure 3.** Difference in the temporally averaged ensemble mean of 2m temperature (in K, upper row), the ratio between the temporally averaged ensemble standard deviations on annual (in %, middle row) and monthly (in %, lower row) timescales between the different sets in epoch 1a.

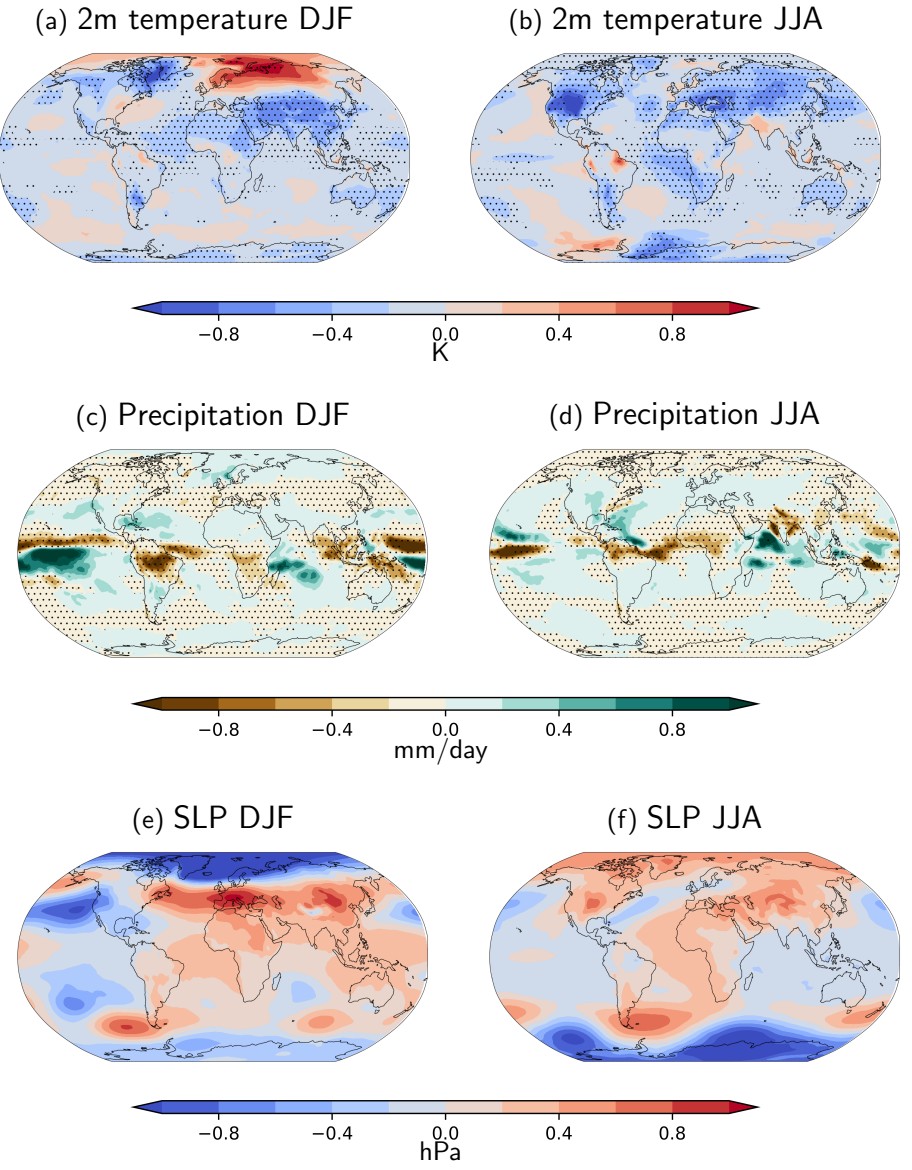

**Figure 4.** Response to volcanic forcing. Difference between the first winter (DJF, left column) and summer (JJA, right column) after an eruption for 2m temperature (top), precipitation (middle) and sea level pressure (bottom) and the same quantity avaraged over the 5 previous summers/winters. The resulting response was then averaged over 15 maior eruptions (same as in Fischer et al. (2007), table 1 therein.) and across all ensemble members. Only significant values are shown, i.e. grid points where the response is outside the 5 to 95 percentile range of 1000 surrogates with each surrogate being created by picking 15 random years and then computing the averaged difference between these 15 random years w.r.t. the 5 years before.

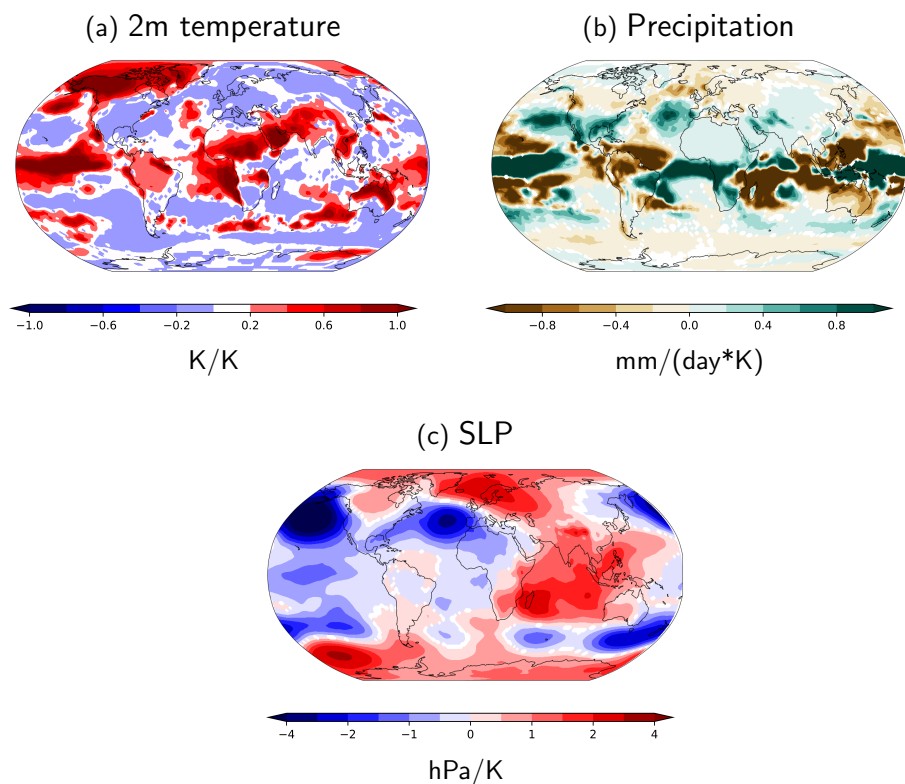

**Figure 5.** Atmosperic response to ENSO. Regression of winter (DJF) means of (a) 2m temperature, (b) precipitation, and (c) sea level pressure on the Nino 3.4 index for all sets of epoche 1a. Values where the correlation is not significant at the 1% confidence level are masked.

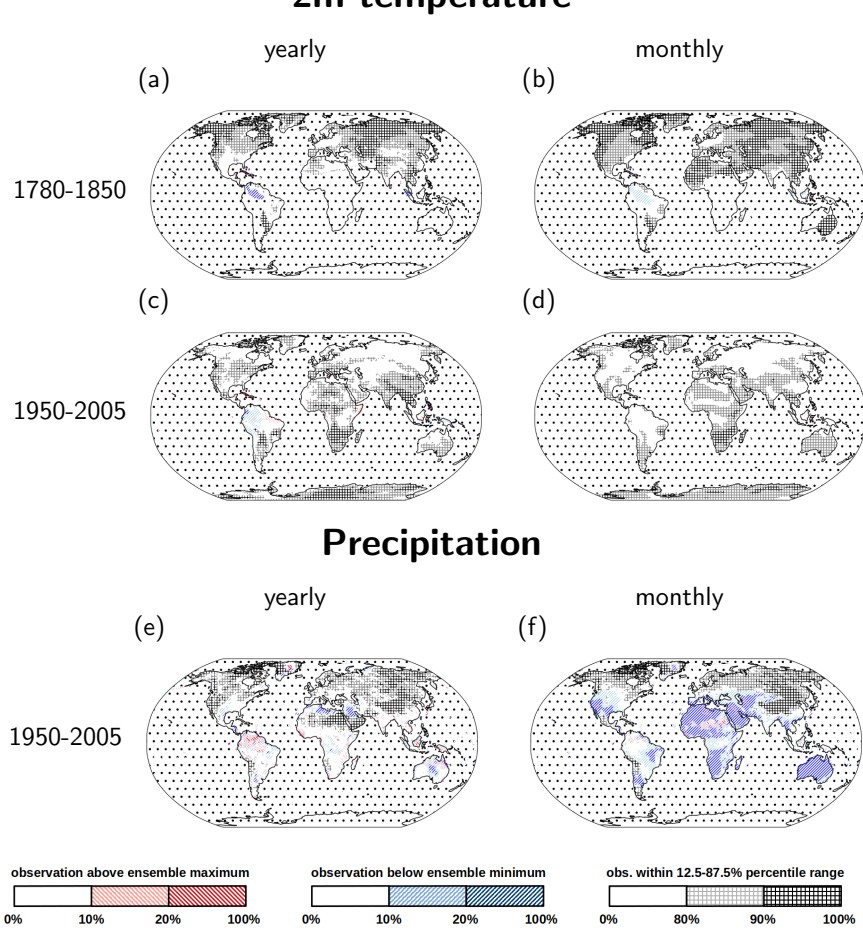

## 2m temperature

## Precipitation

**Figure 6.** Ability of ModE-Sim to capture internal variability for 2m temperature in the period 1780 to 1850 (upper row) and 1950 to 2005 (middle row) and precipitation (lower row) on yearly (left) and monthly (right) timescales. Light red (Dark red) shadings indicate regions where Berkeley Earth (for temperature) respectively GPPC (for precipitation) observations lie below the ensemble maximum of the for more than 10% (20%) of the time steps, light blue and dark blue shadings indicate regions where the observations are below the ensemble minimum accordingly. The grey hatching indicates regions where the ensemble overestimates internal variability, i.e. where more than 80 (light grey hatching) respectively 90 (black hatching) % of the time steps fall within the 12.5 to 87.5 percentile range. Stippling indicates regions where observations are available for less than 10% of the time steps. For details on the section refer to 3.4.

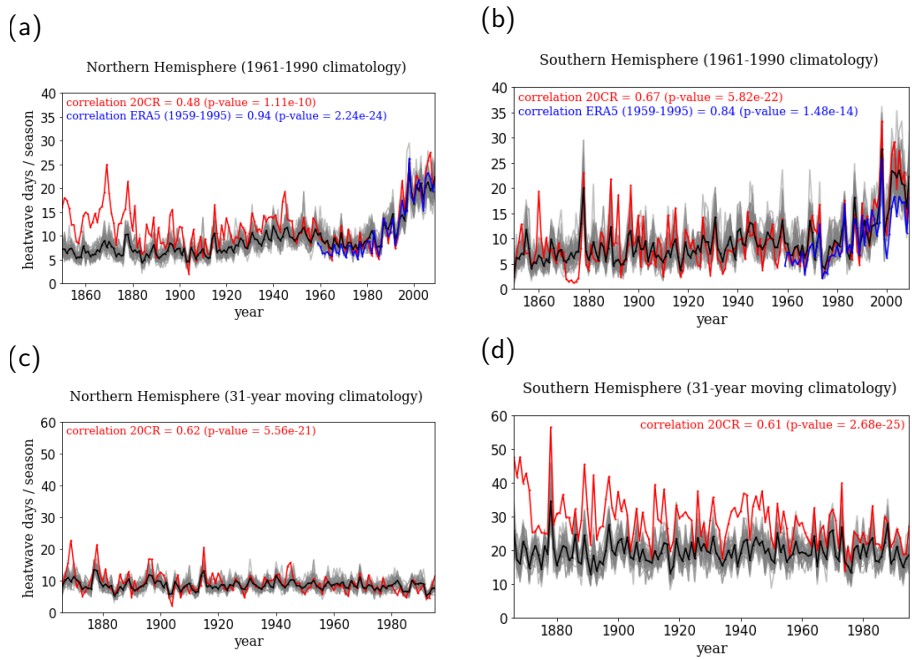

**Figure 7.** Ability of ModE-Sim to capture late 19th and 20th heat waves for (a) Northern hemisphere (10°N to 80°N) in boreal summer (May to September) and (b) Southern hemisphere (65°S-0°) in austral summer (November to March). Shown is the number of heatwave days per season defined as days when 2m temperature exceeds the 90th percentile of the 1961 to 1990 reference period. Light colours show the individual ModE-Sim ensemble members, the black line the ModE-Sim ensemble mean, the blue line ERA5 and the red line the 20CRv3 ensemble mean. (c) and (d) same as (a) and (b), but with using the 90th percentile of a 31-year running climatology.