# Peer review of "ModE-Sim - A medium size AGCM ensemble to study climate variability during the modern era (1420 to 2009)"

_EGUsphere, 2023_

## Author Response (AR1)

**Response to reviewer comments**

**Response to RC1 (https://doi.org/10.5194/egusphere-2023-209-RC1)**

Summary: The manuscript presents a new dataset of climate simulations spanning approximately five hundred years, generated by the global atmospheric model of the Max-Blank Institute. This ensemble of simulations incorporates variations in external forcing and estimated sea-surface temperature and sea ice cover. The authors assess the realism of these ensembles in reproducing the mean climate, climate variations, and the frequency of heat waves. The manuscript is generally well-written, with clear figures and simulation descriptions. However, there are some concerns with specific sections, such as the one discussing heat waves. Additionally, there are some general comments for the authors to consider in the revised version.

We would like to thank reviewer #1 for taking the time to review the manuscript and for the constructive comments. In the following, we provide a point-by-point response to the individual items that were brought up.

Main points:

1) The manuscript should emphasize that these ensembles cannot capture the internal climate variability originating in the ocean. This is mentioned in the text, but it remains somehow hidden, and at some stages, it may be misleading. This point should be highlighted prominently, even in the abstract and title, as the simulations are driven on observed or reconstructed sea surface temperatures without including the possible internal variability.

Thanks for the comment. We have added text in the abstract (ll. 2-5) and the experiment description (ll. 74-78) to clarify this point. We decided to leave the title, as we would prefer to keep it consistent with the name of the dataset which is already published at WDCC and should not be changed anymore, since DOIs were already assigned to it.

ll. 2-5: *"The ensemble uses prescribed sea surface in its LR version (T63/approx. 1.8°horizontal with 47 vertical levels)that covers temperatures, sea ice and radiative forcings that reflect observed values while accounting in uncertainties in the ocean conditions and the timing and strength of volcanic eruptions"*

ll.74-74: *"As a main difference, in contrast to the past 2k simulations that use the coupled version of the model, here we use the stand-alone atmospheric component of the model with prescribed SST and sea ice. This setup induces that the ensemble spread does reflect internal variability in the ocean, but ties the ocean to observed conditions. However, as there is high uncertainty in the SSTs, particularly in the early period, we account for these uncertainties by using ensembles of SSTs with an individual realization of the SST making the ocean forcing for each ensemble member of our simulations."*

2) The section on heat waves needs improvement.

The section on heat waves have undergone a comprehensive revision. In the following we will respond to the individual points brought up by the reviewer concerning that section:

The section's title is misleading, as it generally refers to extreme events, not specifically heat waves.

We have changed the section title to be more precise on the content of that section. New section title:

*"Heat waves as an example to demonstrate ModE-Sim's ability to simulate extreme events"*

The definition of heat waves should be provided in the main text rather than in the caption of Figure 7.

We have added a more extended explanation of the methodology for the heat wave part in the main text:

*"For our analysis we computed the number of heatwaves per season for the model simulations and, as a reference, for 20CRv3 and ERA5 reanalysis. Based on daily 2m temperature data we defined heatwaves as those days on which temperature exceeds the 90th percentile of a reference period at the according grid point. We used two different definitions of the reference period: In the first approach the percentiles were computed from the 1961 to 1990 period. This fixed climatology approach has the advantage that the threshold for a day becoming labeled as a heat wave day does not change during time. The obvious disadvantage is that the number of heat wave days is likely to increase with rising mean temperature. Therefore we additionally provide a second version of the analysis where the percentiles are computed from a 31-year running window. This adapts the temperature threshold for heatwaves to the anthropogenic warming and heatwaves are restricted to days that are extremely warm w.r.t. other days in the according 31-year window. To correct for model biases the thresholds were computed separately for each dataset. To compare the datasets we computed the spearman rank correlation between the model and the reanalysis and according p-values."*

The underestimation of heat wave frequency and intensity before 1920 should be acknowledged more explicitly, as it is not a slight discrepancy.

Discrepancies with 20CR are likely a consequence of poor data coverage and high uncertainties in 20CR during that period. We have added an according statement in the text:

*"Discrepancies with 20CR in the Southern hemisphere in the early 20th century are likely a result of missing data and high uncertainties in 20CR during that period."*

The text should avoid suggesting that most heat waves are caused directly by external radiative forcing, as Figure 7 indicates that they are primarily related to mean temperatures.

Whether the detection of heat-waves is sensitive to changing mean temperatures or not depends on the computation of the percentiles. In the previously used version we computed the quantiles based on a fixed 1961 to 1990 reference period. This will indeed make heat waves more frequently occur in the late 20th century.

In the updated version of the manuscript we added two subplots where we used a 31-year moving climatology for the definition of the percentiles to show that the agreement between reanalysis and model simulations does not only result from the anthropogenic warming signal.

[Figure]

**Figure 7.** Ability of ModE-Sim to capture late 19th and 20th heat waves for (a)  Northern hemisphere (10°N to 80°N) in boreal summer (May to September) and (b)  Southern hemisphere (65°S-0°) in austral summer (November to March). Shown is the number of heatwave days per season defined as days when 2m temperature exceeds the 90th percentile of the 1961 to 1990 reference period. Light colours show the individual ModE-Sim ensemble members, the  black line the ModE-Sim ensemble mean, the blue line ERA5 and the red line the 20CRv3 ensemble mean. (c) and (d) same as (a) and (b), but with using the 90th percentile of a 31-year running climatology.

Furthermore we have clarified that forced signal does not necessarily mean a direct response to the radiative forcing, but may also include dynamical components resulting from changed SSTs and/or volcanic forcings:

*"External forcing in this case does not necessarily restrict to a direct radiative response, but may also include dynamical components that are a result of the SST forcing and/or the volcano forcing. High correlations are not only resulting from the strong temperature trend in the late 20th century, but also hold with the 31-year moving climatology approach (lower row of Fig. 7)."*

To compare the model's ability to capture extreme events, model biases should be corrected, and heat waves could be defined separately in the reanalysis and model data. This can also be achieved by defining the 95% percentile for reanalysis and for model data separately.

Definition of the percentiles was done for each dataset separately. We clarified that by adding a statement on that in the updated version of the manuscript:

*"To correct for model biases the thresholds were computed separately for each dataset."*

A reviewer for the companion paper on heat waves pointed out that the previously used pearson correlation coefficients seems inappropriate due to the extreme value distribution of heat waves. We

therefore replaced the pearson correlation coefficients by spearman rank correlation in the updated manuscript.

Note that the 95% in the previous version were a typo and the correct percentile we used as a threshold was actually the 90$^{th}$ percentile.

Specific points:

3. The information about data assimilation and a gridded 3-dimensional dataset may not be relevant enough for this manuscript for inclusion in the abstract.

We have removed this part from the abstract.

4) The sentence claiming the ensemble's capability to capture extreme events (heat waves) needs careful revision based on the aforementioned points.

We have changed the according sentence in the abstract to *"At the example of heat waves we show that the ensemble is even capable of capturing **certain types of** extreme events"* to clarify that the statement might not universally account for all types of extreme events.

5) The phrase "sampling of the supposed true state" is unclear. Clarification is needed, possibly by reformulating the sentence to avoid ambiguity. If by 'true state' the authors mean the mean climate, then this is not a random variable. It is a fixed parameter of the Earth's climate. However, if by 'true state' the authors mean the climatic probability distribution, then the sentence makes sense.

For clarification we have changed this sentence to *"Also, if the ensemble size is large enough and capable of spanning the full range of physically plausible climate states, large ensembles are likely to include realizations that are close to the historically observed climate state within a reasonalbe range of uncertainty."*

6) Section 2.2 should mention the external forcing used for the simulations upfront rather than referring to it later in the text.

We have rephrased section 2.2 to include a statement on forcings at an earlier point. However, we decided to keep the details of the forcings for the individual sets in the according subsections.

7) In line 108, "set 1430-2" should likely be corrected to "set 1430-3."

corrected to 1420-3.

8) " forcing and the ocean boundary conditions can be detected from the subensemble means computed from each 20-member set separately (Fig. 2b & c), indicating that the ensemble size is clearly sufficient to separate forced signals from internal variability'

The distinction between internal atmospheric variability and internal climate variability with oceanic origins is important and should be clarified. The current phrasing might suggest that the ensemble captures all internal climate variability, which is not the case.

We have changed the sentence to *"indicating that the ensemble size is clearly sufficient to separate forced signals from internal variability **of the atmosphere**"*

**Response to RC2 (https://doi.org/10.5194/egusphere-2023-209-RC2)**

The paper reports on the construction of a single climate model ensemble for the period 1420 to 2009. The ensemble members have different initial conditions but also different boundary conditions (SSTs and sea-ice) and forcings. The climate model is a coarse version of the atmospheric model ECHAM6. The experimental design, including the initial conditions and the forcings, is described in detail. This is followed by a description of the model behavior including the response to volcanic forcings, extremes and heat-waves.

While I value the construction of another large single-model ensemble, and this stands out by including variability related to the uncertainty in the forcings, I cannot recommend that the paper is accepted in its present form. This is a result of the major comments mentioned below.

We would like to thank reviewer #2 for taking the time to review the manuscript and for the constructive comments. In the following we provide a point-by-point response to the individual items that were brought up.

Major comments:

1) The paper seems to be put hastily together. The analysis and results in section 3.4 and 3.5 are very briefly discussed and the methods are basically only described in the figure captions. I can understand that the authors do not wish to present a very deep analysis of heatwaves, but what they choose to show should be adequately described. Figure 3 is only mentioned briefly (l110) and the results are not properly described.

Sections 3.4 and 3.5 have been undergone a comprehensive revision. Particularly we now provide an improved description of the methodology.

 *"We apply the method proposed in the latter reference to ModE-Sim, to analyze whether the ability of capturing internal variability also holds in a stand-alone atmospheric mode and for our ensemble with fewer ensemble members that extend further into the past. The method has the strength to evaluates internal variability in a model ensemble without making a priori assumptions. A detailed description of it can be found in the reference, briefly summarized it works as follows: First, one calculates the ratio of timesteps where the observations fall outside the ensemble minimum to ensemble maximum range at the same timestep. If the ensemble captures the spread correctly, then such outlayers should only occur very occassionally (how often exactly depends on the ensemble size; e.g. a 20-member ensemble with realistic internal variability should be likely to capture all events with a 20-year return period, so on average in 5 % of the timesteps an observation should fall outside the ensemble minimum to ensemble maximum range). If the observation tend to be only above (only below) the ensemble maximum (minimum) for an overproportional number of timesteps then this indicates a negative (positive) mean state bias of the model. If outlayers w.r.t. the ensemble min-to-max range occur frequently into both, positive and negative direction, this indicates that the model underestimates observed internal variability.*

*Finally, the method also allows to detect regions where the model overestimates internal variability w.r.t. observations. This is the case if an overproportional amount of observations falls within the center of the ensemble spread. Consistent with Suarez-Gutierez (2021) we here use the 12.5 to 87.5 quantile range, which - if the ensemble spread agrees with observed internal variability - by definition on average should include 75 % of the observations."*

For the revision on the methodology in section 3.5 please refer to reviewer #1's main comment #2

2) Tests of statistical significance are missing throughout the paper. Regions with significant results should be indicated in the figures.

We have now added significance tests to the responses to the SST forcing and to the volcanoe forcings. For the section on heat waves we added p-values for the correlations to fig. 6.

3) I have several comments to the construction of the additional SSTs in Section 2.2.3. First of all the construction is not linear as claimed. A linear model would look like SST_i = 1/3(SST_j + SST_k + SST_l).
Furthermore, I think x and t should be dropped from the formula and it should just be mentioned in the text that the construction is applied simultaneously to grid-points and time.

We have changed the terminus "linear combination" to "recombination" in the updated version of the manuscript. Furhtermore we removed the indices from the formula as recommended by the reviewer

More importantly, the authors should also be aware that their method generates SSTs that do not have a Gaussian distribution. The distribution of a ratio like SST_j/SST_k has heavy tails. It is difficult to see how big this problem is here, but it will also depend of the units you use for temperature (I hope it is Kelvin).

Division of two sets of independent random variables that are both normally distributed should result in a Cauchy distribution that in deed can be heavy-tailed. However, it has to be noted that in our case we divide not two completely independent datasets that only share the same distribution, but different realizations of HadISST2. At least for those time steps and for those grid points where there is low uncertainty in SST, the individual HadISST2 realizations are correlated. The higher the correlation between the two HadISST realizations is, the closer the quotient should be scattered closely around 1.

As a result the additional SST forcings that we gained by recombination are close to be normally distributed. In fig. R2-1, we show the distribution of all monthly anomalies (w.r.t. climatology):

[Figure]

*Figure R2-1: Global distribution of monthly SST anomalies for HadISST and the recombination of HadISST. The red line describes a gaussian distribution fitted to the data.*

To further confirm that the distributions of HadISST and HadISST have similar distributions, we performed additional statistical tests on a gridpoint-by-gridpoint base: A Kolmogorov-Smirnov test and, for higher test strenght, a Cramér-von-Mises test. The tests were done on SST anomalies, to remove the annual cycle and the long-term temperature signal, we subtracted a 31-year running climatology. Fig. R2-2 shows that outside the regions that are affected by sea ice for both tests the null hypothesis ("the samples are taken from the same distribution") can only be rejected for extremely high p-values.

[Figure]

*Figure R2-2: P-values for a Kolmogorov-Smirnov test (left) and a Cramér-von-Mises test (right).*

Minor comments:

Abstract: The data assimilation procedure is mentioned but it is not clear if this is for the future or if it is included in the present paper. Perhaps the abstract could be more informative on the results of the paper.

Following a comment by reviewer R1 we have removed the part about the data assimilation from the abstract.

l16: This refer to single-models ensembles only. Multi-model ensembles also include differences in physics. I think the difference between the two types of ensembles should be stated directly here.

Changed to:

*"The individual realizations of a single-model ensemble can differ either in their boundary conditions, their initial conditions, or in both."*

l22: This sentence is not clear. .. was done by comparison with the statistics .. Comparison with what?

Changed sentence to:

*"Before the widespread use of large ensembles, separating internal variability and external components of climate variability in model simulations usually was done by comparision of the statistics of a transient simulation with the statistics of a control simulation with climatological forcings."*

l43 A time-slice is mentioned but which time-slice.

Changed to:

*"Providing boundary conditions **for the early part of the modern era** is challenging but necessary."*

l46: Perhaps define modern era in the text.

Definition is now given at an earlier occurrence after slightly changing the first sentence of the according paragraph:

*"Here we present Modern Era SIMulations (ModE-Sim), a medium-sized ensemble of simulations with an atmospheric model capturing the period 120 to 2009 (later refered to as the "modern era")"*

l57: 'last': as in latest or final?

Changed to final.

l103: Is EVA defined anywhere?

We have added the definition in the previous sentence:

*"Consistent with the PMIP4 standard setup, these radiative forcings are outputs of the Easy Volcanic Aerosol Model (Toohey et al., 2016, EVA) using..."*

l108: 1430-2 --> 1430-3 ?

corrected to 1420-3

l168: Perhaps the warming over northern Eurasia is connected to a positive NAO. A positive NAO is reported in observational studies the winter after the eruptions (see, e.g., Christiansen 10.1175/2007JCLI1657.1).

We agree with the reviewer that the warming is very likely related to a positive NAO/AO. This is strongly supported by the SLP pattern shown in figure 4e. In the previous version of the manuscript we have claimed that the winter warming is caused by enhanced warm air advection. Even though by stating this we intended to make the link with the pattern resembling a positive AO-like pattern in Figure 4e, we missed to explicitly use the terminus AO, which we now changed in the revised version of the manuscript. Furthermore, we added the reference provided by the reviewer:

*"… This reduces higher latitude wave breaking and hence disturbances of the polar vortex. In ModE-Sim we can find anonamously negative SLP response in the northern polar latitudes in connection with a band of positive SLP anomaly that spans over the North Atlantic and Northern Eurasia. This pattern, also known as the positive phase of the Arctic Oscillation (or North Atlantic Oscillation when restricted only to the Atlantic region) is a known mode of large scale variability in the Northern hemisphere and its excitation in the first winter after volcanic eruptions is supported by observations (Christiansen, 2008; Fischer et al., 2007). A positive AO/NAO in winter typically leads to enhanced advection of marine air to the continent, resulting in the winter warming. While the warming is significant at least in its center, the SLP response itself is not, likely due to the high internal variability in SLP."*

Section 3.3: How does the model reproduce the temporal characteristic of the ENSO? The responses in Fig. 5 seem very spatial extended. The statistical significance is important here.

We have now added significance tests to the responses to the SST forcing and to the volcanoe forcings. For the section on heat waves we added p-values for the correlations to fig. 6.

[Figure]

**Figure 4.** Response to volcanic forcing  Fischer et al. (2007) . Difference between the  first winter (DJF, left column) and summer (JJA, right column) after an eruption for 2m temperature (top), precipitation (middle) and sea level pressure (bottom) and the same quantity avaraged over the 5 previous summers/winters. The resulting response was then averaged over 15 major eruptions (same as in Fischer et al. (2007), table 1 therein.) and across all ensemble members. Only significant values are shown, i.e. grid points where the response is outside the 1 to 99 percentile range of 1000 surrogates with each surrogate being created by picking 15 random years and then computing the averaged difference between these 15 random years w.r.t. the 5 years before.

[Figure]

**Figure 5.** Atmosperic response to ENSO. Regression of winter (DJF) means of (a) 2m temperature, (b) precipitation, and (c) sea level pressure on the Nino 3.4 index for all sets of epoche 1a. Values where the correlation is not significant at the 1% confidence level are masked.

Figure 2: It is hard to see the shading in panels b and c.

We agree that the visibility of the shadings is challenging at low zoom levels. However we had no idea how to change the figure in a way that comparison between the ensemble mean time series in panel a is still possible (this clearly benefits from having the same scale at the time axis). Splitting panels b and c into three subpanels each (one for each subset separately) might help, but then comparison between the individual subsets would be exacerbated. Therefore, we decided to leave subfigures b and c as they are as a trade-off to allow both, comparison between the subsets and with the overall ensemble mean/spread.

Figure 4: The statistical significance describing where this signal is different from zero should be indicated. The caption says 'ensemble mean': I guess this is the mean of the volcanic signal over the ensemble members and not the signal in the ensemble mean temperature. This should be explained better.

We have now added a monte-carlo-based approach for significance in the figure (see above). Averaging was done over both, all volcanoes and all ensemble members. We slightly changed the caption for clarification.

Line 210 and Fig. 6: The description here is so brief that it is almost impossible to understand.

We have extended the description of the method in section 3.4 and added a reference to that section in the figure caption of fig. 6. (see main comment of reviewer R2)

Section 3.5: Again, I find this much too brief to be of any value.

Sections 3.5 has now undergone a comprehensive revision. The explanation of the methodology to identify heat waves was improved. Furthermore, the revised version provides additional plots to show that agreement between reanalysis and model simulations does not only arise from the long-term change in mean temperature.

See also response to reviewer comment R1 for details.